# Notch Signaling Pathway in Tooth Shape Variations throughout Evolution

**DOI:** 10.3390/cells12050761

**Published:** 2023-02-27

**Authors:** Thimios A. Mitsiadis, Pierfrancesco Pagella, Helder Gomes Rodrigues, Alexander Tsouknidas, Liza L. Ramenzoni, Freddy Radtke, Albert Mehl, Laurent Viriot

**Affiliations:** 1Institute of Oral Biology, Center of Dental Medicine, University of Zurich, 8032 Zurich, Switzerland; 2Centre de Recherche en Paléontologie-Paris (CR2P), UMR CNRS 7207, CP38, Muséum National d’Histoire Naturelle, Sorbonne Université, 75005 Paris, France; 3Laboratory for Biomaterials and Computational Mechanics, Department of Mechanical Engineering, University of Western Macedonia, 50100 Kozani, Greece; 4Section of Computerized Restorative Dentistry, Center of Dental Medicine, University of Zurich, 8032 Zurich, Switzerland; 5Ludwig Institute for Cancer Research, University of Lausanne, 1015 Lausanne, Switzerland; 6Laboratoire de Biologie Tissulaire et d’Ingénierie Thérapeutique, Unité Mixte de Recherche 5305, Université Claude Bernard Lyon 1, CNRS, 69367 Lyon, France

**Keywords:** notch signaling, Jagged1, RNA analysis, mouse, human, tooth, dental morphology, evolution, differentiation

## Abstract

Evolutionary changes in vertebrates are linked to genetic alterations that often affect tooth crown shape, which is a criterion of speciation events. The Notch pathway is highly conserved between species and controls morphogenetic processes in most developing organs, including teeth. Epithelial loss of the Notch-ligand Jagged1 in developing mouse molars affects the location, size and interconnections of their cusps that lead to minor tooth crown shape modifications convergent to those observed along Muridae evolution. RNA sequencing analysis revealed that these alterations are due to the modulation of more than 2000 genes and that Notch signaling is a hub for significant morphogenetic networks, such as Wnts and Fibroblast Growth Factors. The modeling of these tooth crown changes in mutant mice, via a three-dimensional metamorphosis approach, allowed prediction of how Jagged1-associated mutations in humans could affect the morphology of their teeth. These results shed new light on Notch/Jagged1-mediated signaling as one of the crucial components for dental variations in evolution.

## 1. Introduction

Identifying the molecular mechanisms that have driven evolutionary changes in tissues and organs is a critical challenge in current biology. Teeth are the most mineralized tissues in vertebrates and therefore constitute the best-preserved part of the skeleton following fossilization. Consequently, studies of mammalian evolution often rely on the analyses of tooth shape, since subtle changes in tooth crown morphology usually constitute a criterion of speciation events. The remarkable diversity of tooth crown shapes results from differences in number, position, arrangement and interrelation of cusps, as well as on the dimensions of the dental crown [1,2]. Variations in these traits reveal a wide range of adaptations that occurred in relation to numerous episodes of diversification over 200 million years of mammal evolutionary history [3,4]. Hence, the study of the genetic regulation of tooth morphology is important to understand the mechanisms underlying changes in tooth crown shape during evolution. In this context, the mouse dentition is one of the most widely used mammalian models in paleo-evo-devo investigations [5,6,7,8,9,10].

Mammals develop species-specific dentitions whose form and function are directly related to the activation of defined epithelial and mesenchymal signals at various locations of the developing tooth germs [11,12]. A combination of key signaling molecules, including Bone Morphogenic Protein (BMP), Fibroblast Growth Factor (FGF), Sonic hedgehog (Shh) and Wnt families, are produced and secreted by the developing dental tissues. These molecules are also expressed during specific stages of odontogenesis in restricted areas of the dental epithelium that represent tooth exclusive signaling centers comparable to those localized in a variety of other developing organs in vertebrates [11,13]. These signals regulate the proliferative activity of epithelial and mesenchymal cells, leading to dental epithelial folding and the formation of cusps, which constitute the earliest developmental sign of species-specific tooth patterning. Spatial arrangement and the interconnection of cusps is closely linked to the diversity and evolution of dietary habits [4,5,13].

The Notch signaling pathway encompasses a group of evolutionary conserved trans-membrane protein receptors known to be involved in tooth formation and morphology [14,15,16,17]. Four mammalian Notch homologues (Notch1, Notch2, Notch3 and Notch4), which interact with five trans-membrane-bound ligands (Jagged1, Jagged2, Delta1, Delta-like3 and Delta-like4), have been identified [18,19,20,21,22,23]. These molecules have been shown to play crucial roles in binary cell-fate decisions mediated by the lateral inductive cell signaling in many developmental systems. In humans, mutations in the *NOTCH1* and *NOTCH3* are associated with a neoplasia (T-cell acute lymphoblastic leukemia lymphoma) and CADASIL (cerebral autosomal dominant arteriopathy with subcortical infarcts and leukoencephalopathy), respectively [24]. *JAGGED1* mutations have been associated with Alagille syndrome, an inherited autosomal dominant disorder that affects the face and various organs and tissues, including liver, heart, axial skeleton and eyes [25,26]. The functional role of Notch molecules has been investigated in detail by their targeted inactivation in mice. *Notch1^−/−^* [27], *Notch2^−/−^* [28], *Jagged1^−/−^* [29], *Jagged2^−/−^* [30] and *Delta1^−/−^* [31] homozygous null mice die either at early embryonic stages or after birth. Notch deregulation in mice severely affects the development of the kidney, heart, intestine, eyes and somites, as well as neurogenesis and angiogenesis [18,27,28,29,30,31]. Components of the Notch pathway are also expressed during odontogenesis and play an essential role in the patterning and formation of dental tissue matrices [15,17,32]. Our previous findings have shown that *Jagged2*-mediated Notch signaling is required for proper tooth morphology [30].

Our previous studies have shown that *Jagged1* is expressed from the very first stages of odontogenesis in the developing dental epithelium [14]. Loss of *Jagged1* is embryonically lethal due to defects in the embryonic and yolk sac vasculature remodeling [29]. Therefore, to investigate the role of this gene in dental epithelium, we used a tissue-specific deletion strategy (*K14Cre*;*Jagged1^fl/fl^*; GenBank accession number AF171092). We examined the effects of *Jagged1* loss in tooth epithelium and its involvement in tooth crown shape modifications.

## 2. Materials and Methods

### 2.1. Animals

Animal housing and experimentation were performed according to the Swiss Animal Welfare Law and in compliance with the regulations of the Cantonal Veterinary Office, Zurich, Switzerland (licenses: 151/2014, 146/2017, 197/17). The animal facility provided standardized housing conditions, with a mean room temperature of 21 ± 1 °C, relative humidity of 50 ± 5%, and 15 complete changes of filtered air per hour (HEPA H14 filter); air pressure was controlled at 50 Pa. The light/dark cycle in the animal rooms was set to a 12 h/12 h cycle (lights on at 07:00 a.m., lights off at 19:00 p.m.) with artificial light of approximately 40 Lux in the cage. The animals had unrestricted access to sterilized drinking water, and ad libitum access to a pelleted and extruded mouse diet in the food hopper (Kliba No. 3436; Provimi Kliba/Granovit AG, Kaiseraugst, Switzerland). Mice were housed in a barrier-protected specific pathogen-free unit and were kept in groups of max. 5 adult mice per cage in standard IVC cages (Allentown Mouse 500; 194 mm × 181 mm × 398 mm, floor area 500 cm^2^; Allentown, NJ, USA) with autoclaved dust-free poplar bedding (JRS GmbH + Co. KG, Rosenberg, Germany). A standard cardboard house (Ketchum Manufacturing, Brockville, ON, Canada) served as a shelter, and tissue papers were provided as nesting material. Additionally, crinklets (SAFE^®^ crinklets natural, JRS GmbH + Co. KG, Rosenberg, Germany) were provided as enrichment and further nesting material. The specific viral, bacterial and parasitic pathogen-free status of the animals was monitored frequently and confirmed according to FELASA guidelines by a sentinel program [33].

*K14Cre;Jagged1^fl/fl^* conditional knockout mice were generated by crossing *K14:Cre* (MGI: 2445832, Tg(KRT14-Cre)1Amc/J(#004782)) and *Jagged1^flox^* (MGI: 3577993, 129/Sv-Jag1<tm1Frad>) [34] mice. The animals were genotyped using the following primers: Cre Fw, 5′-CTG TTT TGC CGG GTC AGA AA-3′; Cre Rv, 5′-CCG GTA TTG AAA CTC CAG CG-3′; Jag1 Fw, 5′-GCA AGT CTG TCT GCT TTC ATC-3′; Jag1 Rv, 5′-AGG TTG GCC ACC TCT AAA TC-3′. The age of embryos was determined according to the vaginal plug (E0.5) and confirmed by morphological criteria. Animals were killed by cervical dislocation and E18.5 embryos were surgically removed and fixed overnight in 4% paraformaldehyde (PFA) in phosphate-buffered saline (PBS), pH 7.4. Newborn and adult animals were sacrificed by intracardiac perfusion with 4% PFA. Heads were then post-fixed in 4% PFA overnight at 4 °C, thoroughly rinsed with PBS, and placed in 70% ethanol.

Embryos and newborn animals were washed in PBS, incubated in sucrose 30%, embedded in Tissue Tek^®^ O.C.T.^TM^ (4583, Sakura, Alphen aan den Rijn, The Netherlands) and serially sectioned at 10 µm.

### 2.2. Dental Nomenclature, Imaging, Measurements and Phenotyping

The dental nomenclature used here is specific to murid rodents: M^n^ refers to the n^th^ upper molar, M_n_ to the n^th^ lower molar and Mn to both the n^th^ upper and lower molars. Cusps are respectively symbolized by a c^n^ and c_n_ for each upper and lower molar. Main cusps are numbered from 1 to 9 in upper molars and from 1 to 7 in lower molars from the mesio-lingual edge to the disto-vestibular edge of the tooth. High quality images of one wild-type (WT) mouse skull (D1404) and of two mutant mice skulls (D1437 and M12) were obtained using X-ray synchrotron microtomography at the European Synchrotron Radiation Facility (ESRF, Grenoble, France), beamline BM5, with a monochromatic beam at energy of 20 keV and using a cubic voxel of 7.45 µm. This method has been proven to be very useful for very precise imaging of small elements as teeth [35]. Three-dimensional renderings were then performed using VG Studio Max 2.0 software. Dental morphological variations in location, size and interconnection of cusps were analyzed in the WT and *K14Cre;Jagged1^fl/fl^* mice. Length (L) and width (W) for each molar were extracted at 0.001 mm using the LAS Core software (Leica^®^, 4132 Muttenz, Switzerland). The Student’s *t*-test and Fischer’s F-test was used on each dental measurement (W, L and d1–d5) for WT and *K14Cre;Jagged1^fl/fl^* mice to check mean and variance equality. In order to quantify the shape variations within M^1^ and M_1_ mesial parts, five other distances (d1–d5) were measured using LAS Core.

Overall tooth shapes were investigated by using an outline analysis. By registering the relative size and position of each cusp, this method appears suitable for tooth shape study. Fourier methods, notably Elliptic Fourier Transform (EFT), allow the description of complex outlines approximating them by a sum of trigonometric functions of decreasing wavelength (i.e., harmonics). x and y coordinates of 64 points equally spaced along dental outline were calculated to quantitatively describe the shape of M^1^ and M_1_. We applied EFTs to these data using EFAwin software (version 11794, New York State University at Stony Brook, NY 11794, USA) [36], extracting Fourier coefficients from the original outline, and normalizing these shape variables. This method considers the separate Fourier decomposition of the incremental change in x and y coordinates as a function of the cumulative length along the outline [37]. For EFT, any harmonic n yields four Fourier coefficients: An and Bn for x, and Cn and Dn for y, which all contribute to describe the initial outline. We retained the first nine harmonics for M^1^ and the first five for M_1_, which represent the best compromise between measurement error and information content for these murine molars [38]. However, the four coefficients of the first harmonic (A1–D1) were not included in the subsequent analyses because they were poorly discriminant and constituted background noise after the standardization step (size and orientation) [38,39].

### 2.3. Statistical Analyses

We performed the Student’s *t*-test and Fischer’s F-test on each dental measurement (W, L and D1–5) for WT and *K14Cre;Jagged1^fl/fl^* mice to respectively check mean and variance equality. A principal component analysis (PCA) allowed the evaluating of a possible outline variation between WT and *K14Cre;Jagged1^fl/fl^* mice. Variables were represented by the coefficients of each harmonics previously selected for M^1^ and M_1_ (respectively 32 and 16). A multivariate analysis of variations (MANOVA) allowed the researching of a potential significant difference between WT and *K14Cre;Jagged1^fl/fl^* mice cohorts. This test included the coordinates of first axes of the PCA for which the sum met at least 95% of the total variation. These data were previously rank transformed since they did not fulfill the required parameters (i.e., normality, homoscedasticity of variances) for such statistical tests [40].

### 2.4. RNA Sequencing

#### 2.4.1. Samples Preparation

Lower molars were dissected from n = 4 E18.5 *K14Cre;Jagged1^fl/fl^* embryos and n = 4 WT littermates. Left and right lower first molars from the same animal were pooled. RNA was isolated using the RNeasy Plus Mini Kit (Qiagen AG, 8634 Hombrechtikon, Switzerland) and subsequently purified by ethanol precipitation.

#### 2.4.2. Library Preparation

The quality of the isolated RNA was determined with a Qubit^®^ (1.0) Fluorometer (Life Technologies, South San Francisco, CA 94080, USA) and a Bioanalyzer 2100 (Agilent, Waldbronn, Germany). Only those samples with a 260 nm/280 nm ratio between 1.8–2.1 and a 28S/18S ratio within 1.5–2 were further processed. The TruSeq RNA Sample Prep Kit v2 (Illumina, Inc., San Diego, CA 92122, USA) was used in the succeeding steps. Briefly, total RNA samples (100–1000 ng) were poly A enriched and then reverse transcribed into double-stranded cDNA. The cDNA samples were fragmented, end repaired and polyadenylated before ligation of TruSeq adapters containing the index for multiplexing Fragments containing TruSeq adapters on both ends were selectively enriched with PCR. The quality and quantity of the enriched libraries were validated using Qubit^®^ (1.0) Fluorometer and the Caliper GX LabChip^®^ GX (Caliper Life Sciences Inc, Hanover, MD 21076, USA). The product is a smear with an average fragment size of approximately 260 bp. The libraries were normalized to 10nM in Tris-Cl 10 mM, pH8.5 with 0.1% Tween 20.

#### 2.4.3. Cluster Generation and Sequencing

The TruSeq PE Cluster Kit HS4000 or TruSeq SR Cluster Kit HS4000 (Illumina, Inc., San Diego, CA 92122, USA) was used for cluster generation using 10 pM of pooled normalized libraries on the cBOT. Sequencing was performed on the Illumina HiSeq 4000 single-end 125 bp using the TruSeq SBS Kit HS4000 (Illumina, Inc., San Diego, CA 92122, USA).

#### 2.4.4. Data Analysis

Reads were quality checked with FastQC. Sequencing adapters were removed with Trimmomatic [41] and reads were hard-trimmed by 5 bases at the 3′ end. Successively, reads at least 20 bases long, and with an overall average phred quality score greater than 10 were aligned to the reference genome and transcriptome of Mus Musculus (FASTA and GTF files, respectively, downloaded from Ensembl, GRCm38) with STAR v2.5.1 [42] with default settings for single-end reads.

Distribution of the reads across genomic isoform expression was quantified using the R package GenomicRanges [43] from Bioconductor Version 3.0. Differentially expressed genes were identified using the R package edgeR [44] from Bioconductor Version 3.0. A gene is marked as DE if it possesses the following characteristics: (i) at least 10 counts in at least half of the samples in one group; (ii) p ≤ 0.05; (iii) fold change ≥ 0.5. Finally, gene sets were used to interrogate the GO Biological Processes database for an exploratory functional analysis. Contingency tables were constructed based on the number of significant and non-significant genes in the categories and we reported statistical significance using Fisher’s exact test.

#### 2.4.5. Adapter Sequences

Oligonucleotide sequences used in the study are listed in the Table 1.

### 2.5. RNA Isolation RT-PCR

For RT-PCR analysis, lower molars were dissected from n = 8 E18.5 *K14Cre;Jagged1^fl/fl^* embryos and n = 8 WT littermates. Left and right lower first molars from the same animal were pooled. RNA was isolated using the RNeasy Plus Mini Kit (Qiagen AG, 8634 Hombrechtikon, Switzerland) and subsequently purified by ethanol precipitation. Reverse transcription of the isolated RNA was performed using the iScript™ cDNA Synthesis Kit and according to the instructions given (Bio-Rad Laboratories, 1785 Cressier, Switzerland). Briefly, 1000 ng of RNA were used for reverse transcription into cDNA. Nuclease-free water was added to add up to a total of 15 μL; 4 μL of 5× iScript reaction mix and 1 μL of iScript reverse transcriptase were added per sample in order to obtain a total volume of 20 μL. The reaction mix was then incubated for 5 min at 25 °C, for 30 min at 42 °C and for 5 min at 85 °C using a Biometra TPersonal Thermocycler (Biometra AG, Göttingen, Germany). The 3-step quantitative real-time PCRs were performed using an Eco RT-PCR System (Illumina Inc., San Diego, CA, USA).

The reaction mix was composed of 5 μL of SYBR^®^ Green PCR Master Mix reverse and forward primers (200 nM), and 2 ng of template cDNA. The thermocycling conditions were 95 °C for 10 min, followed by 40 cycles of 95 °C for 15 s, 55 °C for 30 s and 60 °C for 1 min. Melt curve analysis was performed at 95 °C for 15 s, 55 °C for 15 s and 95 °C for 15 s. Expression levels were calculated by the comparative ΔCt method (2−ΔCt formula), normalizing to the Ct-value of the *36B4* housekeeping gene.

### 2.6. Immunohistochemistry

Cryosections were air dried for 1 h at room temperature, then washed with PBS to remove excess of Tissue Tek^®^ O.C.T.^TM^. Endogenous peroxidases were inhibited by incubating the sections in a solution composed of 3% H_2_O_2_ in Methanol at −20 °C for 20 min. Specimens were then blocked with PBS supplemented with 2% fetal bovine serum and thereafter incubated with primary antibodies for 1 h at room temperature. The following primary antibodies were used: Rabbit anti-Notch1-ICD (1:50, ab8925, Abcam, Cambridge, UK), Rabbit anti-Hes1 (1:50, 19988, Cell Signaling, Danvers, MA, USA), Rabbit anti-Hes5 (1:50, ab194111, Abcam, Cambridge, UK), Rabbit anti-β-catenin (1:50, 8480S, Cell Signaling, Danvers, MA, USA), Rabbit anti-Ki67 (1:100, ab15580, Abcam, Cambridge, UK), Rabbit anti-Amelogenin (1:100, ab153915, Abcam, Cambridge, UK). For negative controls, primary antibody was omitted. The sections were then incubated with a biotinylated secondary antibody (Vector Vectastain ABC kit PK-4001-1; Vector Laboratories LTD, Peterborough, UK). Sections were then incubated with AEC (3-amino-9-ethylcarbazole; AEC HRP substrate Kit—SK4200; Vector Laboratories LTD, Peterborough, UK) to reveal the staining, counterstained with Toluidine Blue, mounted with Glycergel (C0563, Agilent Technologies, Santa Clara, CA, USA) and imaged with a Leica DM6000 light microscope (Leica Microsystems, Schweiz AG, Heerbrugg, Switzerland).

### 2.7. 3D Morphing of Tooth Geometries and Computation/PREDICTION of human Jagged1-Mutant M^1^

The 3D morphing of all tooth geometries was conducted in ANSA by BETA CAE Systems A.G. (CH-6039 Switzerland) and the methodological approach can be broken down into the following 5 steps.

*Tooth surface generation*. Average tooth morphologies were computed for the M^1^ of WT rodent (based on two characteristic teeth), a mutant one (considering nine *Jagged1*-mutant crown morphologies) and a human M^1^ (resulting from 246 molars). This process facilitated the detection and association of the prevalent features, to compute realistic reference models. Especially for the human molar M^1^, a large existing database was available, which was used as the starting point for calculating the respective average tooth morphology. The data set is based on a set of existing impressions of carious-free and intact tooth surfaces of West European young people within the ages from 16 to 20 years. From the impressions, stone replicas were made, and the clinical visible parts of their surfaces were measured with a 3D-scanning device. The resolution of the measuring process was 50 µm × 50 µm (x,y). All data sets were aligned in position and orientation within the same coordinate system: For this, a representative molar with an appropriate orientation was chosen. All other molars were superimposed with this representative molar by a least-square fitting routine (Match3D) [45,46,47]. This guaranteed that all occlusal features had the same orientation. The pattern of cusps and grooves varies considerably across individuals, even though some morphological properties, such as the overall layout of the main cusps and fissures, are shared by all samples. These common features allow to establish correspondence between the scans z(x,y) with a modified optical flow algorithm in an automated procedure [48]. Based on this correspondence, the *x*, *y*, *z*—coordinates can be averaged and result in a typical representation of the human M^1^, which was used further in this process [45,46].

*Pre-processing.* Prior to applying the morphing algorithms to the teeth, proxy geometries were created, by scaling all teeth (mouse WT and mutant M^1^ and human M^1^) to comparable dimensions. Teeth were then aligned based on characteristic morphological patterns (e.g., tooth cusps and grooves). Mouse and human teeth were positioned by a point-to-point spatial relation of their functional features: the 4 cusps of the human M^1^ (protocone, metacone, paracone, hypocone) were aligned to the 4 distal cusps (c^5^, c^6^, c^8^, c^9^) of the mouse M^1^.

*Creation of a dense 3D correspondence (Features mapping).* The non-uniform triangulated meshes of the *.stl files, were used as bi-linear maps for the morphing algorithm, facilitating the determination of a dense correspondence across both source (WT) and target (mutant) tooth geometry. The mouse mutant M^1^ consisted of 249.797 triangular elements whereas the mouse WT M^1^ and human M^1^ consisted of 252.296 and 51.476, respectively. A correspondence was algorithmically established, by identifying the minimum projected distance between each node of the source mesh to a node or an interpolation of multiple nodes on the target geometry. Source nodes that are not paired with correspondences, are handled as “in-between positions” and placed as simple linear interpolations of the vertex positions in the target mesh. The use of these nodes in the mesh triangulation increases the quality of the morph function, despite not being visible on the target.

To provide a reference plane for the translation matrix, pairs of points were placed on the source and target geometry to identify locations that should be in correspondence. These boundary locations were selected below the functional surfaces of the molars (in proximity to the cervical margin line) to ensure the unobstructed morphing of the molar crown morphology. Areas of interest were isolated based on the statistically significant variations found in mouse WT M^1^ vs. mutant M^1^. The computed transformation matrix was then applied to the 3D data set of the human M^1^, to predict the crown morphology in individuals carrying Jagged1 mutations.

*Construction of the 3D morphs*. Polygonal surface approaches, such as 3D morphing, require mapping of the characteristic topological landmarks (as described above), followed by re-meshing of the morphed surfaces to achieve a realistic convergence. The quality-oriented reconstruction of the source model’s grid produces a robust 3D morph.

*Application of the 3D morphs to human molars.* The transformation vectors required to morph the human healthy M^1^ into a M^1^ in individuals carrying Jagged1 mutations were formulated as an algorithmic matrix. This matrix was then stored as a geometry-independent parameter, reflecting the differences of the crown surface between WT and mutant M^1^. Transferring the deformation map of this parameter to the 3D data set of the human M^1^ (average model) facilitates the computation of the M^1^ crown morphology, which is expected to be representative both in terms of morphology and size of the human mutant M^1^.

## 3. Results

### 3.1. Morphological Modifications in the Crown of K14Cre;Jagged1^fl/fl^ Molars

The teeth of the *K14Cre;Jagged1^fl/fl^* mice exhibited crown morphological differences when compared to the teeth of WT mice, mainly regarding the location and interconnection of the cusps (Figure 1Aa; red arrows).

The most striking variations on the upper molars (i.e., M^1^, M^2^, M^3^) were observed in their first cusp (c^1^ cusp). In the WT mice, the c^1^-c^2^ connection of the M^1^ seen in the side-view was usually achieved by a high V-shaped crest, which contrasts to the frequently observed (in approximately 60% of analyzed samples) low U-shaped profile in M^1^ of the *K14Cre;Jagged1^fl/fl^* mice (Figure 1Aa,b; red lines). This implies that the c^1^-c^2^ connection was weak to absent in the *K14Cre;Jagged1^fl/fl^* mutant mice (Table 2).

This variation was confirmed by height measurements of the c^1^–c^2^ connection (Figure 1B; d1), which were significantly different between the two cohorts. In addition, these measurements revealed that the spacing between c^1^ and c^2^ cusps was significantly greater in the M^1^ of mutant mice (Figure 1B; d2). The c^1^ cusp of the M^1^ had a more linguo-distal position in the majority of the *K14Cre;Jagged1^fl/fl^* mice, and this feature explained its greater spacing from c^2^. On the M^2^ of mutant mice, the vestibular extension of the cusp c^1^ spur was absent in about 60% of specimens (Figure 1Aa; Table 2). The c^1^ cusp of the M^3^ was sometimes reduced and tended to merge with the c^4^ cusp in 30% of the *K14Cre;Jagged1^fl/fl^* specimens (Figure 1Aa; Table 2). The central cusps (c^5^ and c^8^) of the mutant M^1^ and M^2^ had a rather angular or even pointed mesial edge, while this part was always smooth in the molars of WT mice (Figure 1Aa). This latter morphotype occurred in approximately 60% of the upper molars of *K14Cre;Jagged1^fl/fl^* mice (Table 2), but was not always simultaneously present on each cusp. The lower molars (i.e., M_1_, M_2_, M_3_) of the mutant mice showed little variation compared to the upper molars, yet the c_1_ and c_2_ cusps of the mutant M_1_ appeared closer to each other compared to the WT M_1_ (Figure 1Aa). Three measurements revealed significant differences in d3-d5 mean lengths (Table 3) and confirmed that c_1_ and c_2_ cusps tended to partly merge in the M_1_ of mutant mice, while the cusp c_1_ was less protruding.

The first two axes of the PCA on M^1^ outlines were poorly discriminant because WT and *K14Cre;Jagged1^fl/fl^* specimens plotted together, but the third axis was more discriminant although morphospaces partly overlapped (Figure 1Ca). The main dental trend expressed on the third component was the variation of the cusp c^1^ location, according to extreme outlines on the negative and positive sides. This cusp was indeed located in a more distal-lingual position in the extreme outline of the negative side where a majority of *K14Cre;Jagged1^fl/fl^* specimens plots. The results of the principal component analysis (PCA) were also confirmed by a multivariate analysis of variations (MANOVA), pointing out a significant difference in M^1^ outline between WT and *K14Cre;Jagged1^fl/fl^* mice (Table 4).

Contrary to M^1^, the first component of the PCA on M_1_ outlines represented the most discriminant axis, and the second one was more discriminant than the third (Figure 1Ca). Nonetheless, the M_1_ morphospaces overlapped as well. The main difference between WT and *K14Cre;Jagged1^fl/fl^* was linked to the mesial protrusion of the c_1_ cusps, which was less important on extreme outlines from the negative to the positive side of the first component. A significant difference between the two cohorts was also displayed by the MANOVA (Table 4). Mean sizes (L and W) of all molars were significantly lower in *K14Cre;Jagged1^fl/fl^* mice compared to WT specimens (Figure 1Cb; *t*-test), while there was no significant difference concerning the variances (Figure 1Cb; Fischer’s F-test).

### 3.2. Transcriptome Modifications in K14Cre;Jagged1^fl/fl^ Molars

To understand how the epithelial deletion of *Jagged1* modulates the whole dental developmental program we compared the transcriptome of first molars isolated from E18 *K14Cre;Jagged1^fl/fl^* embryos and WT littermates (Figure 2A). RNA sequencing analysis showed a significant change in the expression of more than 2000 genes upon epithelial deletion of *Jagged1* (Figure 2A,B). These genes encode diverse categories of proteins, including proteins linked to binding, catalytic and transported activities (Figure 2C). Unbiased Gene Ontology (GO) Enrichment Analysis identified several networks affected by the loss of Jagged1 (Figure 2D).

We found a significant upregulation of genes involved in mineralization, autophagy, ion transport and cell cycle arrest (Figure 2D), which are critical processes for enamel and dentin formation [49,50]. Several genes encoding for proteins necessary for enamel formation, such as Amelx, Ambn, Enam, Mmp20, and Klk4 were upregulated in *K14Cre;Jagged1^fl/fl^* molars (Figure 2E). However, these results varied between the *K14Cre;Jagged1^fl/fl^* molars (Figure 2F). In addition, the expression of mesenchymal genes was also affected in *K14Cre;Jagged1^fl/fl^* molars. For example, genes associated with odontoblast differentiation such as *Dspp* and *Dmp1* were upregulated, while *Pax9*, *Barx1*, *Dlx1* and *Dlx2* were significantly downregulated in the mutant molars (Figure 2D,F). Furthermore, we showed that genes encoding for molecules of the Wnt and FGF signaling pathways were affected (Figure 2E,F). Concerning the Wnt pathway, we observed major downregulation of genes encoding Wnt11, Wnt10b, Wnt9b ligands, Frizzled receptors and Tcf transcription factors (Figure 2C–E), while the genes encoding Wnt3a, Wnt6, Wnt7a and Wnt10a ligands, as well as Apc were upregulated (Figure 2E). Regarding the FGF pathway, *Spry1*, *Spry2* and *Spry4* were significantly downregulated (Figure 2E,F and Figure 3).

Notch pathway members, such as *Jagged1*, *Hes5*, *Hes6*, *Lfng* and *Maml2* were downregulated, while others, such as *Hey1* and *Dll4* were upregulated (Figure 2D,E and Figure 3). Apart from the upregulation of several cell cycle arrest-related genes, no significant alterations were observed in specific networks linked to cell proliferation events (Figure 2Da).

### 3.3. The Expression Patterns of Genes and Proteins Is Affected in K14Cre;Jagged1^fl/fl^ Molars

To validate the above results and analyze whether the site of expression of different genes and proteins was affected by *Jagged1* epithelial deletion, we performed real-time PCR (RT-PCR) analysis, in situ hybridization, and immunohistochemistry on cryosections of E18 *K14Cre;Jagged1^fl/fl^* mouse embryos (Figure 4).

RT-PCR analysis confirmed *Jag1* downregulation as well as *Notch1*, *Notch2* and *Hes5* downregulation in *K14Cre;Jagged1^fl/fl^* molars (Figure 4A). We further demonstrated overexpression of ameloblast (e.g., *Amelx* and *Ambn*) and odontoblast (e.g., *Dspp* and *Dmp1*) differentiation markers in *K14Cre;Jagged1^fl/fl^* molars (Figure 4A). RT-PCR also confirmed alterations in the expression of genes coding for components of the Wnt signaling pathway, namely the upregulation of *Wnt10a* and *Apc*, and downregulation of *Lef1*, *Tcf3* and *Fzd2* (Figure 4A).

In situ hybridization analysis in E18 WT molars showed intense *Jagged1* expression in cells of the inner dental epithelium, which was drastically reduced in this cell layer in E18 mutant molars (Figure 4B). Although strong *Notch1* expression *was observed* in all cells of the stratum intermedium in WT molars, its expression was downregulated in cells located at the cusp territories of *K14Cre;Jagged1^fl/fl^* molars (Figure 4B). Similarly, *Notch2* expression was downregulated in cells of the outer enamel epithelium and stellate reticulum of E18 *mutant* molars, when compared to the gene expression in WT molars (Figure 4B). The expression of *Jagged2* in inner dental epithelial cells was comparable between the mutant and WT molars (Figure 4B), thus suggesting a *Jagged2* compensation for the loss of *Jagged1* that explains the mild morphological changes observed in the mutant molars.

Immunohistochemistry confirmed the deregulation of the Notch signaling pathway in the dental epithelium of *K14Cre;Jagged1^fl/fl^* molars. Labeling against the active form of Notch1 (i.e., Notch1 intracellular domain) showed its distribution in cells of the stratum intermedium of E18 WT molars, contrasting the lack of immunostaining in mutant molars (Figure 4C). Similarly, although a strong Hes1 staining was observed in the stratum intermedium of WT molars, in the cusps of the mutant molars the staining was not obvious (Figure 4C). Hes5 staining was mainly detected in inner dental epithelium (preameloblasts), stellate reticulum and odontoblasts of the E18 WT molars (Figure 4C). Comparison of Hes5 immunoreactivities between WT and mutant molars indicated a slight reduction in the staining in the inner dental epithelium and odontoblasts in the cusp areas (Figure 4C). Amelogenin was distributed in preameloblasts located at the cusps of the WT molars, as well as in parts of the dental pulp (Figure 4C). A similar pattern for Amelogenin was observed in the *K14Cre;Jagged1^fl/fl^* molars, although the staining appeared more expanded when compared to that of WT teeth (Figure 4C). Staining with the proliferation marker Ki67 showed mitotic activity in few cells of the stratum intermedium located in the cusps of the WT molars (Figure 4C). In mutant teeth, increased Ki67 immunoreactivity was observed in cells of the stratum intermedium and some preameloblasts (Figure 4C).

### 3.4. A Computation Mathematical Three-Dimensional (3D) Model Predicting the Shape of Human Molars upon Jagged1 Mutation

The understanding of the molecular bases underlying fine variations in mouse dental morphology can help us to unravel the common factors that shaped teeth throughout evolution. Therefore, we investigated how the deregulation of Jagged1 function would affect the morphology of human molars. The statistically significant variations between WT and mutant rodent teeth can be mathematically applied to the human teeth, thus allowing to predict the crown morphology in humans carrying Jagged1 mutations. For building this model, we opted for M^1^ where the most accentuated modifications were observed in the mutant mice. A polygonal surface technique (3D direct morphing, using ANSA by BETA CAE Systems S.A.) was applied to compute a translation matrix. The transformation vectors of the translation matrix contain all the important information that would allow approximation of the WT M^1^ crown morphology to the *K14Cre;Jagged1^fl/fl^* one (Figure 4Aa,b), while providing an overview of the required node displacements (Figure 5Ac). These node displacements can be described by transformation vectors (Figure 5Ad), which can be transferred to the human M^1^. Consequently, we superimposed WT mouse M^1^ and human M^1^ morphologies (Figure 5B). Despite the fact that significant variations in mouse M^1^ also exist in its mesial part that is not comparable to the human M^1^, an analogue was drawn to the remaining variations in the distal part (containing the c^5^, c^6^, c^8^ and c^9^) of the mouse M^1^. The transformation matrix of each of these four cusps in the mouse M^1^ was applied to the 3D data set of the average human M^1^ (Figure 5C), to predict tooth morphology in individuals carrying Jagged1 mutations (Figure 5Cb,Da–d). The most significant morphological alterations were observed in the occlusal surface of M^1^, with a mesial-lingual/palatal shift of the hypocone and a mesial displacement of the metacone (arrows in Figure 5Cb,Da,b). The morphologies of the protocone and paracone were only marginally affected by the mutation. The distal ridge of the mutant M^1^ is predicted to have a notable deeper and narrower profile (Figure 5Dd) when compared to a typical M^1^ morphology (Figure 5Dc). The distal-lingual/palatal groove was also computed as slightly widened in its mesial part (Figure 5Db), while no significant changes were observed in the vestibular surface of the mutant M^1^.

## 4. Discussion

The characteristics of the *K14Cre;Jagged1^fl/fl^* mouse molars provide relevant information in terms of tooth crown microevolution in rodents. The evolution of the rodent dentition is well understood from an abundant and intensively studied fossil record [1,2,8,51]. The transition from ancestral to descendant species within most evolutionary lineages usually involves minor modifications of tooth shapes. The subtle changes in the interconnections between cusps, their varying frequency and the overlapping tooth size range are reminiscent of cases of dental variations among muroid rodents. Pleistocene and Holocene species of field mice (*Apodemus*) have been shown to possess upper and lower first molars with variable interconnections between their first and second cusps, and these variations occur both at intraspecific and close interspecific levels [52]. Variations in the interrelationships between mesial cusps of the first upper and lower molars have also been described in different Miocene species of *Democricetodon* [53] and *Megacricetodon* [54]. Similar cases of interspecific overall morphological similarities of first molars are also described in early murids, such as *Progonomys* species, showing only subtle differences [55]. In most of these examples, the tooth size range remains nearly identical.

The loss of *Jagged1* function in dental epithelium can thus cause morphological changes in the same cusps that are modified during murid evolution. Consequently, an alteration in *Jagged1* expression might be responsible for subtle tooth crown modifications that underlie processes of dental phenotype adaptation during population splitting or even speciation events over evolution. Tooth morphogenesis requires the orchestrated activity of several molecular networks, and the Notch pathway plays a pivotal role in this process, acting as a hub regulator of main signaling pathways [56]. *Jagged1* epithelial loss induces important modifications of numerous genes that are involved in an important and diverse number of signaling pathways. Loss of *Jagged1* in mice did not result in dramatic changes in the activity of these signaling networks, but rather fine and discrete modulations, which still significantly impacted crown morphology. This indicates that Jagged1 fine tunes a sensitive balance between various gene networks whose interactions govern organogenesis. Disturbance of this genetic equilibrium might ultimately lead to the generation of subtle morphological modifications in most tissues and organs.

The perturbed expression levels of Wnt signaling molecules are expected due to the well-known genetic interaction with the Notch signaling pathway [57,58,59]. The Wnt signaling is a powerful morphogen [60] involved in the establishment of planar cell polarity (PCP) that plays a crucial role in tissue patterning during all developmental stages [61]. It has been already shown that Wnt signaling is fundamental for tooth development since mutations in this pathway are associated with dental anomalies [62,63,64,65,66,67]. Significant alterations in the Wnt signaling pathway in the dental epithelium of Jagged1 mutants suggest that tooth crown morphological changes are mostly due to PCP dysfunction. This hypothesis is further reinforced by the deregulation of genes that are crucial for the establishment of PCP, such as *Vangl2* and *Celsr1* [61], as well as by the altered expression patterns of *Notch1* and *Notch2* in mutant teeth. Misexpression of the Notch receptors and ligands in dental epithelium, in combination with the deregulation of various Notch downstream effectors, will also affect the fate of dental epithelial cells [18]. *Jagged1* mutation in dental epithelium also induces downregulation of the FGF inhibitors *Spry1, Spry2* and *Spry4*, which are important genes affecting tooth number [68] and enamel formation [69].

No substantial changes occurred in genes related to cell proliferation, despite the known correlation between Notch signaling and cell proliferation [70]. However, the upregulation of genes associated with cell cycle arrest, together with the upregulation of genes involved in ameloblast (e.g., *Amelx*, *Ambn, Enam, Mmp20, and Klk4*) and odontoblast (e.g., *Dspp* and *Dmp1*) differentiation, indicates premature cytodifferentiation events in mutant molars. As a consequence, the accelerated deposition of enamel and dentin by the ameloblasts and odontoblasts, respectively, will prematurely stabilize tooth crown shape in the mutants. Advanced cytodifferentiation in mutant molars could thus lead to the observed reduced size and subtle morphological alterations of their crown. Such dental morphology does not resemble any particular murid rodent or any specific dental pattern that can be found in the evolution of Muridae. This is different from changes observed upon *Fgf3* deregulation in mice that greatly affected tooth morphology [6]. Therefore, the present results should be evaluated from a microevolutionary perspective rather than from a macroevolutionary viewpoint. Indeed, the trends observed here for some characters (e.g., first and second cusp connection, first and second cusp complex), their variable frequency and the overlapping of both size range and shape reminds us of cases of wild sibling species of Muroidea [71,72]. It was shown that two extant field mice (i.e., *Apodemus sylvaticus* and *Apodemus flavicollis*) share dental morphologies and nearly the same size ranges [71]. Moreover, the extinct “hamster-like” species, *Eucricetodon asiaticus* and *Eucricetodon jilantaiensis*, from the Oligocene of Ulantatal (Inner Mongolia, China), displayed some overall shape variations even closer to the *K14Cre;Jagged1^fl/fl^* and WT case. The size range of these species is indeed nearly identical, they share intermediary dental morphotypes, their global shapes overlap and their main size distinction relies on mean L/W ratio [72]. Consequently, the modulation of Jagged1-mediated signaling could be the driving force for minor dental modifications that contribute to speciation phenomena, despite the reported negative effect of a haploinsufficiency of this gene in some organs [26,34,73,74].

The extent to which genetic changes contribute to morphological variations in human dentition remains a fundamental question in evolutionary biology. Despite the obvious differences between mouse and human molars (in cusp number, size and interconnections as well as in relative dental proportions), different studies have indicated some possible equivalences [75]. The creation of a model, based on mammalian evolution of common ancestral origin [76] and using computing 3D metamorphosis techniques [77], predicted meaningful morphological changes in the crown of molars in humans carrying *Jagged1* mutations.

## 5. Conclusions

In conclusion, our results suggest that the systematic effect of *Jagged1* deletion provides a basis for dental variations in evolution. Mutation morphologically similar to *Jagged1* deletion sheds a completely new light on the mechanisms of development, which could reconcile microevolutionary and macroevolutionary processes.

## 6. Limitations of the Study

Here, we have demonstrated that the deletion of *Jagged1* from the dental epithelium results in changes to the morphology of the tooth crown, and we suggest the involvement of the Notch signaling pathway in subtle evolutionary tooth shape changes. Although this is a simple, plausible hypothesis, our data do not provide direct evidence of this event. We also propose that these shape modifications are due to the modulation of significant morphogenetic networks that are affected upon *Jagged1* deletion. However, it is challenging to analyze separately and in detail the cascade of the molecular events leading to tooth morphological modifications. Finally, we predicted tooth shape alterations in individual *Jagged1* mutations, using a computing mathematical model. Nevertheless, morphological variations in the teeth of humans carrying *Jagged1* mutations remain to be demonstrated.

## Figures and Tables

**Figure 1 cells-12-00761-f001:**
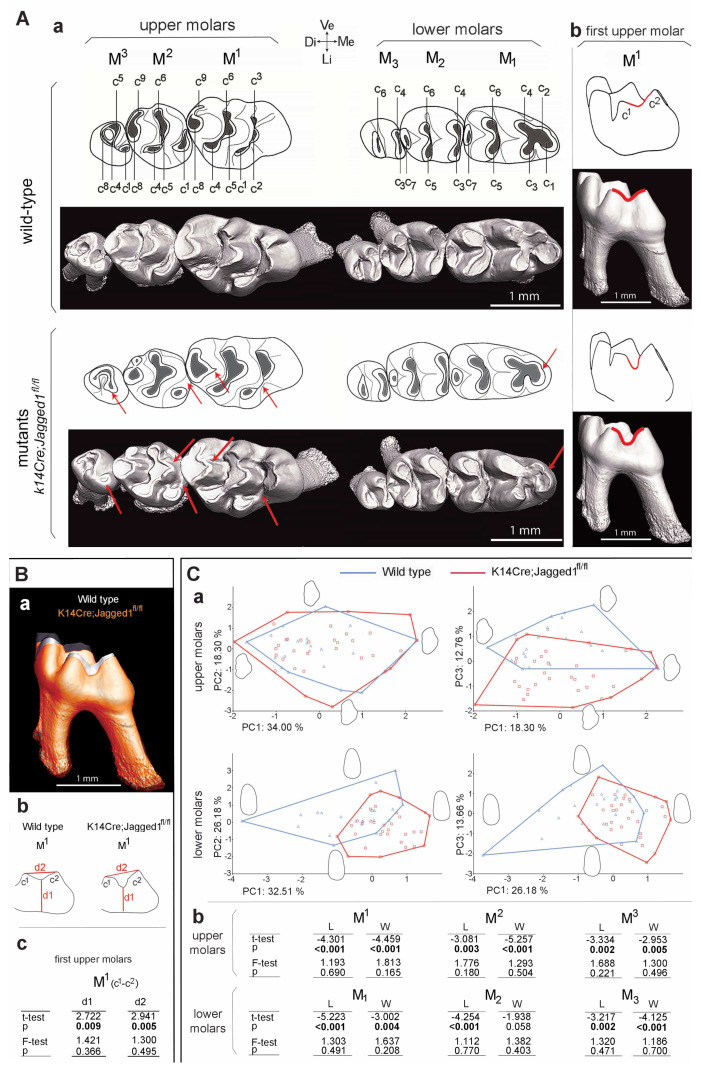
**Epithelial loss of *Jagged1* affects the molar dental morphology.** (**A**) MicroCT analysis showing the morphology of upper and lower molars of wild-type (WT) and *K14Cre;Jagged1fl/fl* mutant mice (both occlusal and side views). (**Aa**) Morphological variations in location, size and connection of the cusps in molars of the *K14Cre;Jagged1fl/fl* mice are indicated by red arrows. (**Ab**) WT first upper molar (M^1^) showing a V-shape c^1^-c^2^ connection profile, and mutant M^1^ showing a U-shape profile (red lines). In mutant mice, the lower first molars (M_1_) show in the mesial part mild fusion between the c_1_ and c_2_ cusps. (**B**) Comparison of the M^1^ crowns in WT and Jagged2 mutants. (**Ba**) MicroCT analysis showing smaller crowns in mutant M^1^ (orange pseudo color) when compared to the crowns of WT M^1^ (grey color). (**Bb**,**c**) Comparison of the distance between the inter-cusp groove and the basis of the crown (d1) and the distance between c^1^ and c^2^ (d2) in M^1^ of WT and mutants. (**C**) Principal component analyses comparing the variation of dental outline of M^1^ and M_1_ between WT (blue color) and *K14Cre;Jagged1fl/fl* (red color) mice. Abbreviations: M, mesial part; D, distal part; V, vestibular part; L, lingual part.

**Figure 2 cells-12-00761-f002:**
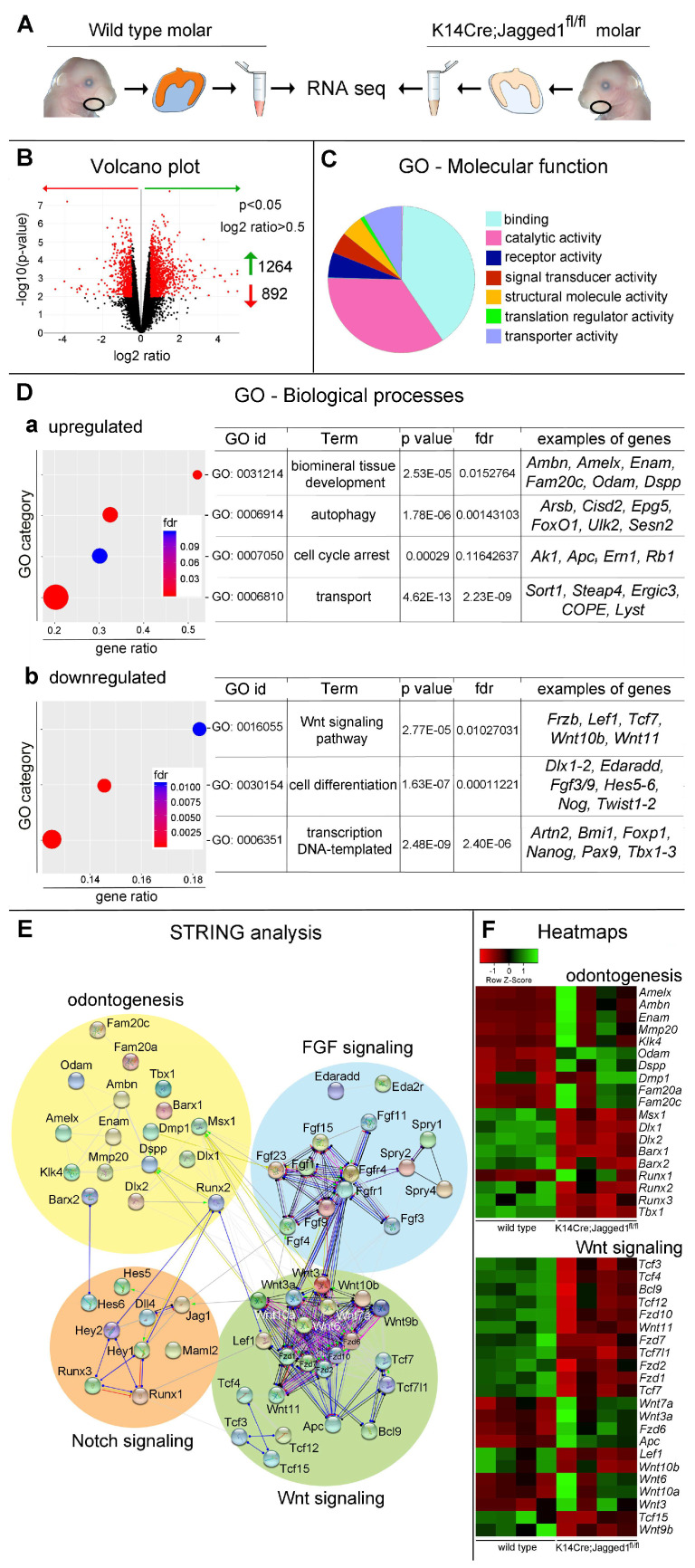
**RNA sequencing analysis of first molars isolated from E18.5 WT and *K14Cre;Jagged1^fl/fl^* mouse embryos.** (**A**) Experimental approach. Molars were isolated from E18.5 *K14Cre;Jagged1^fl/fl^* embryos and WT littermates. Their transcriptome was isolated via mRNA sequencing. (**B**) Volcano plot showing downregulated (red arrow) and upregulated (green arrow) genes in *K14Cre;Jagged1^fl/fl^* mutant mice compared to the WT littermates. (**C**) Pie chart representation of unbiased Gene Ontology Analysis—Molecular Function. (**D**) Hypergeometric representation of unbiased GO—Biological Processes analysis showing the upregulated (**Da**) and downregulated (**Db**) genes in the *K14Cre;Jagged1^fl/fl^* molars. (**E**) STRING analysis of selected protein networks whose gene expression was found to be significantly affected by *Jagged1* deletion in the dental epithelium. (**F**) Heatmaps showing the main deregulated genes involved in odontogenesis and in the Wnt signaling pathway in *K14Cre;Jagged1^fl/fl^* molars.

**Figure 3 cells-12-00761-f003:**
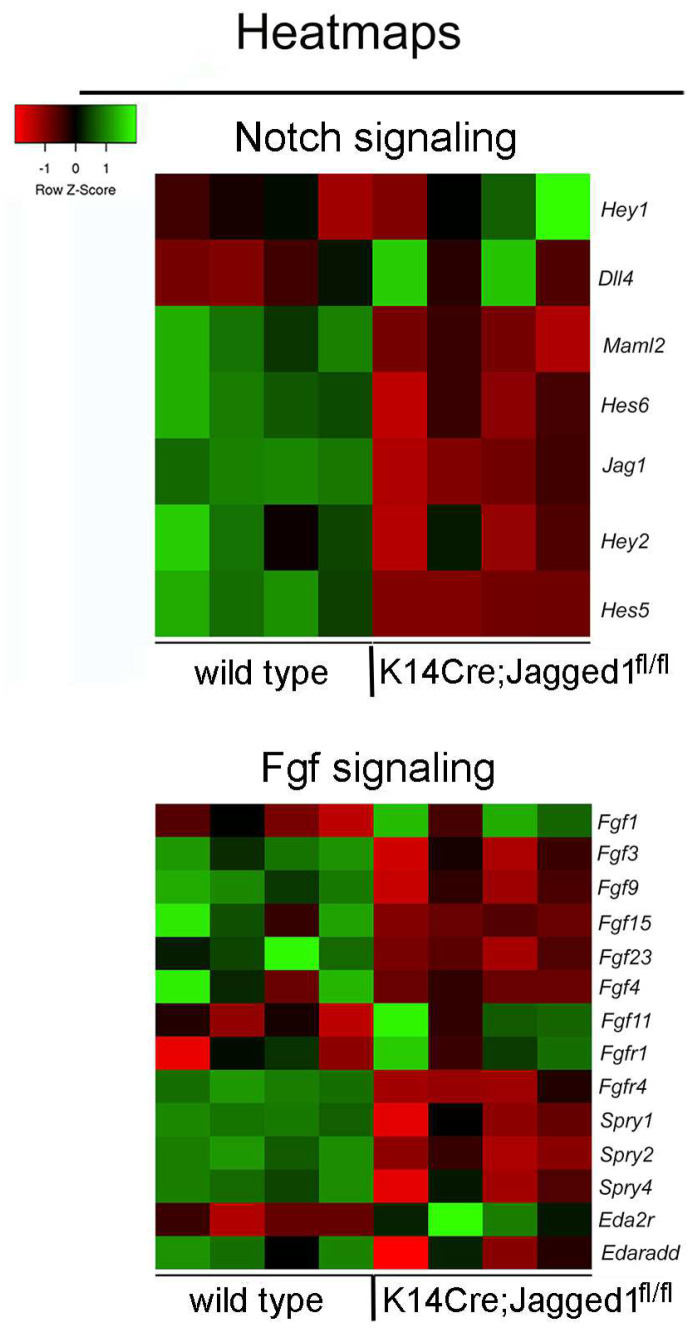
**Heatmaps showing the main deregulated genes involved in the Notch and FGF signaling pathways in *K14Cre;Jagged1^fl/fl^* molars**.

**Figure 4 cells-12-00761-f004:**
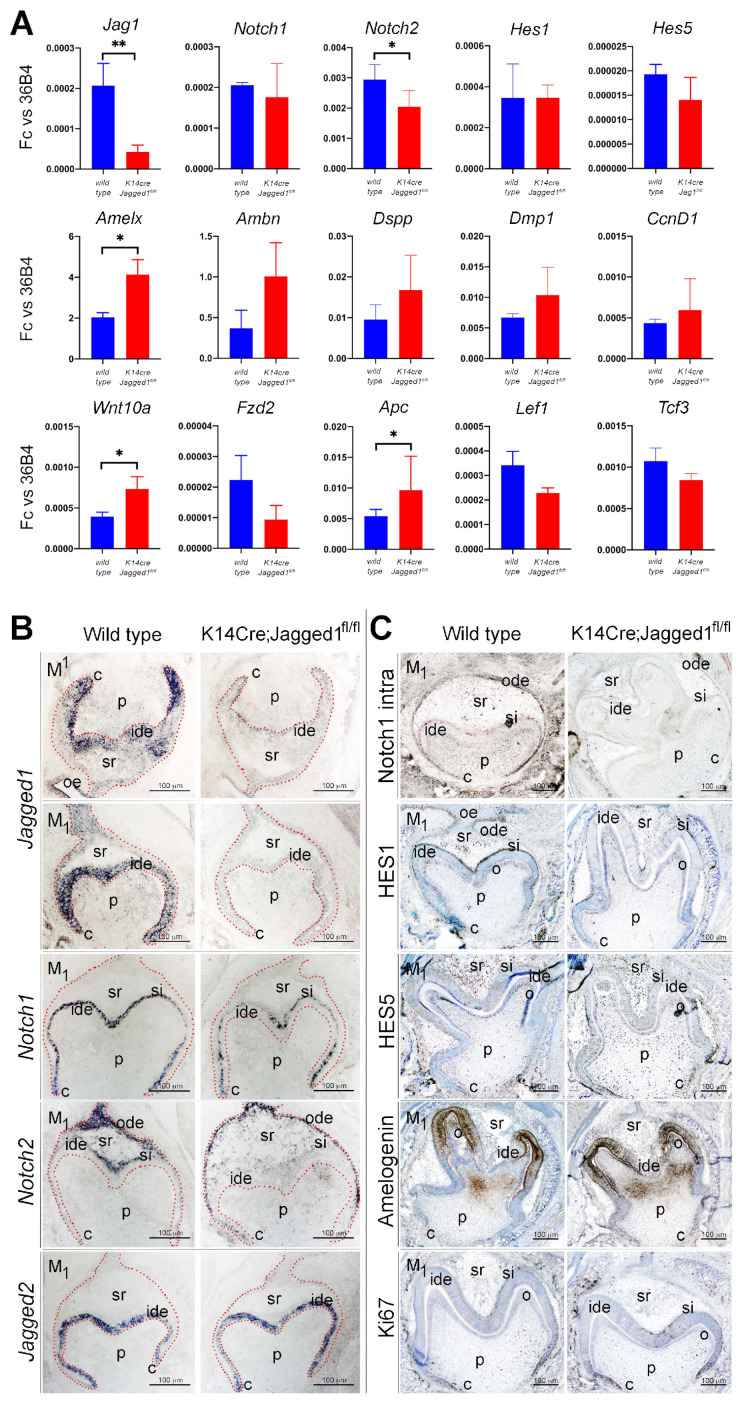
**Expression analysis of genes and proteins in E18.5 WT and *K14Cre;Jagged1^fl/fl^* teeth.** (**A**) RT-PCR analysis of the expression of genes coding for components of the Notch pathway (i.e., *Jag1, Notch1*, *Notch2*, *Hes1*, *Hes5*), Wnt signaling pathway (i.e., *Wnt10A*, *Fzd2*, *Apc*, *Lef1*, *Tcf3*), various ameloblast (i.e., *Amelx*, *Ambn*) and odontoblast (i.e., *Dspp*, *Dmp1*) differentiation markers, and a proliferation marker (i.e., *CcnD1*), in both WT (blue color) and mutant (red color) first molars. Statistical analysis used Student’s t-test (significant difference at *, *p* < 0.05; **, *p* < 0.01). (**B**) *In situ* hybridization showing alterations in the expression pattern of *Jagged1*, *Notch1*, *Notch2* and *Jagged2* in mutant first molars when compared to WT ones. (**C**) Immunohistochemistry showing modifications in the distribution pattern of the active form of Notch1 (Notch1 intra), Hes1 and Hes5 proteins, the enamel-specific Amelogenin protein and the nuclear Ki67 protein (cell proliferation marker) in *K14Cre;Jagged1^fl/fl^* teeth when compared to WT teeth. Scale bars 100 µm.

**Figure 5 cells-12-00761-f005:**
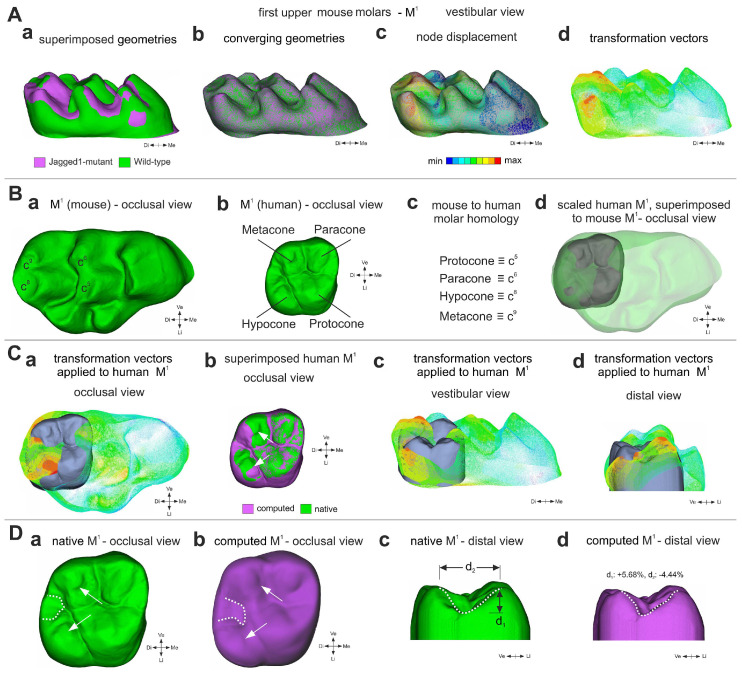
**Prediction of tooth crown morphologies in humans carrying *Jagged1* mutations.** (**A**) Compared morphologies of WT and *K14Cre;Jagged1^fl/fl^* M^1^. (**Aa**) Pre-processing (scaling, positioning and alignment) of proxy geometries, illustrating the affinity of source and target surfaces of rodent M^1^. Green color represents WT teeth, violet color indicated mutant teeth. (**Ab**) Conversion of the morphed WT M^1^, approximating the *K14Cre;Jagged1^fl/fl^* M^1^ geometry. (**Ac**) Magnitude of nodes movement required for the approximation of the WT M^1^ geometry to the *K14Cre;Jagged1^fl/fl^* M^1^ geometry. (**Ad**) Indicative view of the transformation vectors from WT M^1^ to *K14Cre;Jagged1^fl/fl^* M^1^. (**B**) Tooth homology between mouse and human M^1^ molars. (**Ba**) Occlusal view of mouse WT M^1^ indicating the comparable to the human M^1^ cusps. (**Bb**) Occlusal view of human M^1^ indicating the four cusps. (**Bc**) Nomenclature homologation in-between the human M^1^ and mouse M^1^ cusps. (**Bd**) Scaled human M^1^ superimposed on WT mouse M^1^. (**C**) Transformation vectors of the morphing process representing the magnitude of the source nodes’ (WT M^1^) movement during their approximation of the *K14Cre;Jagged1^fl/fl^* M^1^ geometry. (**Ca**,**c**,**d**) Transformation vectors superimposed on human M^1^. (**Cb**) Superimposed native (green color) and computed (violet color) human M^1^. (**D**) Comparison of the native human M^1^ and predicted human mutant M^1^. (**Da**,**b**) Native (green color in (**Da**)) and computed (violet color in (**Db**)) human M^1^ indicating statistically significant changes in cusps (arrows) and the inter-cusp space (dotted lines). Occlusal view. (**Dc**,**d**) Native (green color in (**Dc**)) and computed (violet color in (**Dd**)) human M^1^ showing a notable deeper and narrower profile of the distal ridge in mutant teeth (dotted lines in (**Dd**)) compared to the native teeth (dotted lines in (**Dc**)). Distal view.

**Table 1 cells-12-00761-t001:** **Oligonucleotide sequences for TruSeq™ RNA and DNA Sample Prep Kits**.

Adapter	Sequence
TruSeq Universal Adapter	5′ AATGATACGGCGACCACCGAGATCTACACTCTTTCCCTACACGACGCTCTTCCGATCT
TruSeq Adapter, Index 1	5′ GATCGGAAGAGCACACGTCTGAACTCCAGTCACATCACGATCTCGTATGCCGTCTTCTGCTTG
TruSeq Adapter, Index 2	5′ GATCGGAAGAGCACACGTCTGAACTCCAGTCACCGATGTATCTCGTATGCCGTCTTCTGCTTG
TruSeq Adapter, Index 3	5′ GATCGGAAGAGCACACGTCTGAACTCCAGTCACTTAGGCATCTCGTATGCCGTCTTCTGCTTG
TruSeq Adapter, Index 4	5′ GATCGGAAGAGCACACGTCTGAACTCCAGTCACTGACCAATCTCGTATGCCGTCTTCTGCTTG
TruSeq Adapter, Index 5	5′ GATCGGAAGAGCACACGTCTGAACTCCAGTCACACAGTGATCTCGTATGCCGTCTTCTGCTTG
TruSeq Adapter, Index 6	5′ GATCGGAAGAGCACACGTCTGAACTCCAGTCACGCCAATATCTCGTATGCCGTCTTCTGCTTG
TruSeq Adapter, Index 7	5′ GATCGGAAGAGCACACGTCTGAACTCCAGTCACCAGATCATCTCGTATGCCGTCTTCTGCTTG
TruSeq Adapter, Index 8	5′ GATCGGAAGAGCACACGTCTGAACTCCAGTCACACTTGAATCTCGTATGCCGTCTTCTGCTTG
TruSeq Adapter, Index 9	5′ GATCGGAAGAGCACACGTCTGAACTCCAGTCACGATCAGATCTCGTATGCCGTCTTCTGCTTG
TruSeq Adapter, Index 10	5′ GATCGGAAGAGCACACGTCTGAACTCCAGTCACTAGCTTATCTCGTATGCCGTCTTCTGCTTG
TruSeq Adapter, Index 11	5′ GATCGGAAGAGCACACGTCTGAACTCCAGTCACGGCTACATCTCGTATGCCGTCTTCTGCTTG
TruSeq Adapter, Index 12	5′ GATCGGAAGAGCACACGTCTGAACTCCAGTCACCTTGTAATCTCGTATGCCGTCTTCTGCTTG
TruSeq Adapter, Index 13	5′ GATCGGAAGAGCACACGTCTGAACTCCAGTCACAGTCAACAATCTCGTATGCCGTCTTCTGCTTG
TruSeq Adapter, Index 14	5′ GATCGGAAGAGCACACGTCTGAACTCCAGTCACAGTTCCGTATCTCGTATGCCGTCTTCTGCTTG
TruSeq Adapter, Index 15	5′ GATCGGAAGAGCACACGTCTGAACTCCAGTCACATGTCAGAATCTCGTATGCCGTCTTCTGCTTG
TruSeq Adapter, Index 16	5′ GATCGGAAGAGCACACGTCTGAACTCCAGTCACCCGTCCCGATCTCGTATGCCGTCTTCTGCTTG
TruSeq Adapter, Index 18 4	5′ GATCGGAAGAGCACACGTCTGAACTCCAGTCACGTCCGCACATCTCGTATGCCGTCTTCTGCTTG
TruSeq Adapter, Index 19	5′ GATCGGAAGAGCACACGTCTGAACTCCAGTCACGTGAAACGATCTCGTATGCCGTCTTCTGCTTG
TruSeq Adapter, Index 20	5′ GATCGGAAGAGCACACGTCTGAACTCCAGTCACGTGGCCTTATCTCGTATGCCGTCTTCTGCTTG
TruSeq Adapter, Index 21	5′ GATCGGAAGAGCACACGTCTGAACTCCAGTCACGTTTCGGAATCTCGTATGCCGTCTTCTGCTTG
TruSeq Adapter, Index 22	5′ GATCGGAAGAGCACACGTCTGAACTCCAGTCACCGTACGTAATCTCGTATGCCGTCTTCTGCTTG
TruSeq Adapter, Index 23	5′ GATCGGAAGAGCACACGTCTGAACTCCAGTCACGAGTGGATATCTCGTATGCCGTCTTCTGCTTG
TruSeq Adapter, Index 25	5′ GATCGGAAGAGCACACGTCTGAACTCCAGTCACACTGATATATCTCGTATGCCGTCTTCTGCTTG
TruSeq Adapter, Index 27	5′ GATCGGAAGAGCACACGTCTGAACTCCAGTCACATTCCTTTATCTCGTATGCCGTCTTCTGCTTG

**Table 2 cells-12-00761-t002:** **Morphological variations in the upper molars of *K14Cre;Jagged1^fl/fl^* mice.** On the first upper molars (M^1^) of wild-type (WT) mice, the interconnection of the first (c1) and second (c2) cusps has a V-shape profile, while this profile is often U-shaped (60%) in *K14Cre;Jagged1^fl/fl^* M^1^.

Specimen	Cohort	M^1^		M^2^		M^3^		Upper Molars	
		Right	Left	Right	Left	Right	Left	Right	Left
		c1-c2	c1-c2	c1	c1	c1	c1	Anormal cusp	Anormal Cusp
D1399	*Jag1^fl/fl^*	V-shape	V-shape	-	-	reduced	-	-	-
D1400	*Jag1^fl/fl^*	U-shape	U-shape	no spur	-	-	-	-	c8M2
D1401	*Jag1^fl/fl^*	V-shape	V-shape	-	-	reduced	-	c5M2	c5M1, c8M2
D1402	*Jag1^fl/fl^*	U-shape	U-shape	-	-	-	-	c8M2	c8M2
D1404	*Jag1^fl/fl^*	U-shape	U-shape	no spur	no spur	-	reduced	c5M2	c8M2
D1405	*Jag1^fl/fl^*	U-shape	V-shape	no spur	no spur	-	-	c8M2	-
D1406	*Jag1^fl/fl^*	V-shape	V-shape	no spur	no spur	-	-	c8M2	c8M2
D1407	*Jag1^fl/fl^*	V-shape	V-shape	no spur	-	-	-	-	-
D1408	*Jag1^fl/fl^*	U-shape	U-shape	-	-	-	-	c8M2	-
D1409	*Jag1^fl/fl^*	U-shape	V-shape	-	no spur	-	-	c8M1-2	c8M2
D1413	*Jag1^fl/fl^*	U-shape	U-shape	no spur	no spur	-	-	-	-
D1414	*Jag1^fl/fl^*	U-shape	V-shape	no spur	no spur	-	-	-	c8M1-2
D1415	*Jag1^fl/fl^*	U-shape	U-shape	no spur	no spur	-	-	c8M1-2	c8M1-2
D1433	*Jag1^fl/fl^*	U-shape	V-shape	-	no spur	-	reduced	c8M2	c8M1-2
D1434	*Jag1^fl/fl^*	U-shape	U-shape	no spur	no spur	-	-	-	-
D1435	*Jag1^fl/fl^*	V-shape	V-shape	no spur	no spur	-	-	c8M2	c8M1-2
D1436	*Jag1^fl/fl^*	U-shape	U-shape	no spur	no spur	-	-	-	c8M2
D1439	*Jag1^fl/fl^*	U-shape	V-shape	-	no spur	-	-	c8M2	-
D1440	*Jag1^fl/fl^*	V-shape	U-shape	no spur	no spur	-	-	c8M2	c5-8M2
D1443	*Jag1^fl/fl^*	V-shape	U-shape	no spur	no spur	reduced	reduced	-	c8M1
D1444	*Jag1^fl/fl^*	V-shape	V-shape	no spur	no spur	reduced	reduced	c5M2	-
M8	*Jag1^fl/fl^*	U-shape	U-shape	-	no spur	big	-	c8M2	c8M3
M9	*Jag1^fl/fl^*	V-shape	U-shape	no spur	no spur	reduced	reduced	-	c8M2
M10	*Jag1^fl/fl^*	V-shape	V-shape	no spur	-	reduced	reduced	c8M2	c8M1-2
M11	*Jag1^fl/fl^*	U-shape	U-shape	-	no spur	reduced	reduced	-	c8M1
M12	*Jag1^fl/fl^*	U-shape	U-shape	-	-	reduced	reduced	-	-
M15	*Jag1^fl/fl^*	V-shape	V-shape	-	-	-	-	-	-
M16	*Jag1^fl/fl^*	V-shape	U-shape	no spur	no spur	reduced	reduced	c8M2	c5M1, c8M2
M18	*Jag1^fl/fl^*	U-shape	U-shape	no spur	-	-	reduced	c8M2	c5-8M2
M19	*Jag1^fl/fl^*	U-shape	U-shape	-	no spur	-	-	c8M2	c8M1
M21	*Jag1^fl/fl^*	U-shape	V-shape	no spur	-	-	-	c8M1-2	c8M1-2
M22	*Jag1^fl/fl^*	U-shape	U-shape	-	-	-	-	-	-
M23	*Jag1^fl/fl^*	U-shape	V-shape	-	no spur	-	reduced	-	c8M1-2
D1398	WT	V-shape	V-shape	-	-	-	-	-	-
D1403	WT	V-shape	V-shape	-	-	-	-	-	-
D1410	WT	V-shape	V-shape	-	-	-	-	-	-
D1411	WT	V-shape	V-shape	-	-	-	-	-	-
D1412	WT	V-shape	V-shape	-	-	-	-	-	-
D1426	WT	V-shape	V-shape	-	-	-	-	-	-
D1427	WT	V-shape	V-shape	-	-	-	-	-	-
D1428	WT	V-shape	V-shape	-	-	-	-	-	-
D1429	WT	V-shape	V-shape	-	-	-	-	-	-
D1430	WT	V-shape	V-shape	-	-	-	-	-	-
D1431	WT	V-shape	V-shape	-	-	-	-	-	-
D1432	WT	V-shape	V-shape	-	-	-	-	-	-
D1437	WT	V-shape	V-shape	-	-	-	-	-	-
D1438	WT	V-shape	V-shape	-	-	-	-	-	-
D1441	WT	V-shape	V-shape	-	-	-	-	-	-
D1442	WT	V-shape	V-shape	-	-	-	-	-	-
M13	WT	V-shape	V-shape	-	-	-	-	-	-
M14	WT	V-shape	V-shape	-	-	-	-	-	-
M17	WT	V-shape	V-shape	-	-	-	-	-	-
M20	WT	V-shape	V-shape	-	-	-	-	-	-
M24	WT	V-shape	V-shape	-	-	-	-	-	-

**Table 3 cells-12-00761-t003:** **Morphological variations in the lower molars of *K14Cre;Jagged1^fl/fl^* mice.** Differences in d3-d5 mean lengths indicating that the c1 and c2 cusps are partly merged in the first lower molars (M_1_) of *K14Cre;Jagged1^fl/fl^* mice.

	M_1_ (c_1_-c_2_)		
	d3	d4	d5
*t*-test	−3.121	−4.307	−3.387
*p*	0.003	<0.001	0.001
F-test	1.564	1.101	1.534
*p*	0.296	0.837	0.317

**Table 4 cells-12-00761-t004:** **Variations in the first molars outline of *K14Cre;Jagged1^fl/fl^* mice.** Multivariate analysis of variations (MANOVA) indicates significant differences in the first molars outline between WT and *K14Cre;Jagged1^fl/fl^* mice.

MANOVA	Test	Value	F	d.f.	d.f. (Error)	*p*
M^1^	Wilk	0.340	7.427	11	42	<0.001
M_1_	Wilk	0.557	4.470	8	45	<0.001

## Data Availability

The accession number for all sequencing data reported in this paper is GEO: GSE198920; the reviewers can access the data using the token: mdmzaeqcbholbuf. All data will be made publicly available upon publication of the manuscript.

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
