# Peer review of "Notch Signaling Pathway in Tooth Shape Variations throughout Evolution"

_cells, 2023, doi:10.3390/cells12050761_

Round 1
Reviewer 1 Report
The work submitted by Mitsiadis and collaborators is the continuation of the previous wor focused in the study of Jagged1 during the very first stages of odontogenesis in the developing dental epithelium. In the present manuscript they explore the characterization of Epithelial loss of the Notch ligand Jagged1 in developing mouse molars. They use RNA sequencing analysis to reveal the induced alterations in morphogenetic gene networks of the WNT and FGF signaling pathways. The results put the relevance of the evolutionary changes to understand dental variations in vertebrates through evolution.
The manuscript is well written and the introduction adequate. Material and methods are provided with sufficient detail to be reproduced, however I suggest to put the Adapter sequences in a table to make it easier for the reader. The results are clearly presented, after RNAseq there are subsequent validations of expression at both mRNA (RT-PCR) and protein (immunohistochemistry). Finally the authors use a computation mathematical 3D model to predict the shape of human molars upon Jagged1 mutation and clearly states the limitations of the present study. The amount of work and data provided is overwhelming and objectively I can't find anything to criticize since rational suggestions mean work for two or three additional articles and will not change substantially the main message of the present work.
Author Response
We thank the reviewer for the excellent comments concerning our work and manuscript.
We provided a table (new Table 1) with the adaptor sequences, making the follow up of the paper and results easier for the readers.
We have also added references concerning tooth evolution/morphology and Notch signaling.
Reviewer 2 Report
Thanks for your valuable study. The authors had a study on Notch signaling pathway in tooth shape variations throughout evolution. It is valuble in its field. The introduction provides sufficient background and include all relevant references, however, I think it is so long and did not contain recent and relevant references.
the research design appropriate.
The methodology is complete.
The figures and tables are good. however, the quality of figures are so low.
Generally I enjoyed the manuscript.
Author Response
We thank the reviewer for the encouraging comments and appreciation of our paper.
We have supplemented the revised manuscript with new and relevant references. The quality of the original images is high, however the low quality isdue to the conversion of the word file to PDF. Higher resolution images are provided to the journal.
Thank you again for the positive feedback.
Reviewer 3 Report
The topic of the present original study, investigating the effects of Jagged1 loss in tooth epithelium and its involvement in tooth crown shape modifications, is interesting.
Reported findings currently presented may pave the way for further investigations in humans and may be relevant from the perspective of a better comprehension of odontogenesis and tooth anomalies.
The manuscript is very well-organized and written.
Materials and Methods and Results sections are clearly presented. The discussion section is well structured.
Minor suggestions are given in the pdf file.

Author Response
We thank the reviewer for the encouraging comments and appreciation of our paper.
We have supplemented the revised manuscript with references about tooth morphology. We have also edited the text according to the comment.
Thank you again for the positive feedback.